# Epistemically unwarranted beliefs scale, development and evidence of validity in the Chilean population

Rodrigo Ferrer-Urbina[1], Herman Elgueta[2‡], Marcos Carmona-Halty[1‡], Geraldy Sepúlveda-Paez[3‡], Karina Alarcón-Castillo[1]*

**1** Escuela de Psicología y Filosofía, Universidad de Tarapacá, Arica, Chile, **2** Departamento de Psicología, Universidad de Magallanes, Punta Arenas, Chile, **3** Departamento de Psicobiología y Metodología en Ciencias del Comportamiento, Universidad Complutense de Madrid, Madrid, España

☯ These authors contributed equally to this work.
‡ These authors also contributed equally to this work.
* karina.alarcon.castillo@gmail.com

## Abstract

The study of epistemically unwarranted beliefs (EUB) (i.e., paranormal, pseudoscientific and conspiracy beliefs) has become relevant due to the negative effects they have produced on people's health, as evidenced in the covid-19 pandemic. However, there is no instrument with appropriate and updated validity evidence for its evaluation in Latin American people. Because of this, the present study aims to develop a brief scale to analyze general epistemically unwarranted beliefs that do not depend on local factors. A total of 634 adults from five Chilean cities participated in the study of whom 93.8% (n = 575) were university students. Exploratory and confirmatory factor analyses revealed that the final structure of the Epistemically Unwarranted Beliefs Scale (EUBS) considers 9 items with three related factors. In addition, results showed good internal consistency (CFI > .95; TLI > .95; RMSEA < .07), gender invariance, and evidence of validity based on the inverse relation with the cognitive reflection test and the relationship with sociodemographic variables (i.e., gender, political orientation, and religious orientation). Finally, implications for the theoretical construct and possible limitations of the scale are discussed.

## Introduction

Belief in ghosts, in magnetotherapy or in great conspiracies for the total domination of humanity are beliefs that, despite being far from modern rationality, are widely accepted by a significant part of the population. This type of beliefs are known as epistemically unwarranted beliefs (EUB; or epistemically suspect beliefs), defined, in general terms, as those beliefs that lack empirical support so far [1–3]. In other words, these beliefs do not consider the totality of evidence and knowledge available to those seeking to understand reality at any given time. Although many of these

**Data availability statement:** The data underlying the results presented in the study are available from https://osf.io/f2tmb.

**Funding:** This work was supported by the Universidad de Tarapacá (https://www.uta.cl), Proyecto de Fortalecimiento de Grupos de Investigación #3798–24 (MC-H), and by the Agencia Nacional de Investigación y Desarrollo (https://anid.cl) of the Government of Chile, Fondecyt Regular Project N°1220664 (RF-U). The funders had no role in study design, data collection and analysis, decision to publish, or preparation of the manuscript.

**Competing interests:** The authors have declared that no competing interests exist.

beliefs have been in the humanity since ancient times, however the massification of the communication media, particularly with the proliferation of social networks, has greatly increased their speed of propagation and reach [4].

Although some of these beliefs may be innocuous, several negative effects of adhering to them have been evidenced. For example, conspiracy beliefs were negatively related to health protective behavior [5] and enhanced pseudoscientific practices as a form of treatment for covid-19 virus [6]. Pseudoscientific beliefs were negatively related to scientific literacy [7] and to endorsement of alternative medicine medical treatments [8]. Paranormal beliefs were positively related to sleep variables (i.e., sleep paralysis, lucid dreaming, hypnagogic hallucinations) [9] and to cognitive biases (i.e., jumping to conclusions, catastrophizing, dichotomous thinking) [10]. In addition, it has been shown that, in aggregate, EUB are negatively related to cognitive ability [11], and cognitive reflection [12]. Specifically, interventions aimed at developing cognitive reflection have been shown to cause a decrease in EUB [13–15].

Moreover, the endorsement of epistemically unwarranted beliefs (EUB) has been shown to vary according to several sociodemographic and ideological factors. For instance, regarding gender, women tend to score higher than men in general levels of EUB [1]. In terms of ideology, a consistent association has been found between political conservatism and greater endorsement of such beliefs [13,15–17]. Similarly, religiosity also shows a significant relationship, with religious affiliation being positively related to EUBs, while atheism shows a negative association [1,16–18]. Taken together, these findings suggest that both sociocultural identity and personal belief systems play an important role in shaping individuals' susceptibility to endorsing epistemically unwarranted beliefs.

Epistemically unwarranted beliefs are differentiated by domains, being commonly referenced [2]: 1) paranormal beliefs, such as beliefs about physical, biological or psychological phenomena with fundamental characteristics that do not correspond with their ontology (e.g., premonitory thoughts) [19]; 2) pseudoscientific beliefs, such as beliefs in pseudo-theories, i.e., statements that lack scientific evidence, or deny that which has been scientifically proven (e.g., homeopathy) [20]; and 3) conspiracy beliefs, which point to theories about the existence of secret plots that explain social circumstances (e.g., explanations about the attack on the twin towers) [21]. Despite this differentiation by domains, the available evidence shows a strong relationship between them, so that if, for example, a person believes in conspiracy theories, he or she is likely to adhere to paranormal and pseudoscientific beliefs, and vice versa [1,2,22], which makes sense to consider them globally.

Although the evidence supports the cooccurrence of EUB, the measurement instruments that approach them tend to focus on a specific domain, with the exception of the scales proposed by Lobato et al. [2], Dyer y Hall [13] and Huete-Pérez et al. [1]. Lobato et al. [2] formulated the first multidimensional scale of epistemically unwarranted beliefs, which integrates pseudoscientific, paranormal and conspiracy beliefs. This questionnaire was developed based on existing questionnaires. Although it provided evidence of the interrelationship between these beliefs, the proposal did not present robust psychometric evidence, except for some reliability

estimates. Subsequently, Dyer and Hall [13] developed the Inventory of Epistemically Unwarranted Beliefs (IEUB), which consists of 5 dimensions (paranormal, religion, health, extraordinary life forms, conspiracy theories and ghosts), but did not provide evidence of scale reliability and/or validity. Finally, Huete-Pérez et al. [1], developed the Popular epistemically unwarranted belief inventory (PEUBI), with the intention of measuring epistemically unwarranted beliefs in Spain, based on 5 dimensions that emerged from an exploratory factor analysis (superstitions, occultism and pseudoscience, traditional religion, extraordinary life forms and conspiracy theories), which, although it reported robust evidence of validity, the authors emphasize the idiosyncratic nature of the scale.

In Chile, although there are no instruments with evidence of validity that evaluate the EUB, there are several studies that have considered how this type of beliefs affects the Chilean population, showing the relevance of this construct for the local context. For example, Castillo-Riquelme et al. [23], using Epstein's subscales of magical thinking, naive optimism and esoteric thinking [24], showed that paranormal beliefs are positively related to the credibility of fake news. Halpern et al. [25] used the Conspiracy Mentality scale based on the Conspiracy Mentality Questionnaire (CMQ; [26]) and found that conspiracy beliefs explain susceptibility to disinformation. Salazar-Fernandez et al. [27] developed the validation of a scale on conspiracy beliefs regarding the covid-19 vaccine (CBS Scale), demonstrating that conspiracy beliefs are negatively related to beliefs about the effectiveness of covid-19 vaccines. Finally, Armstrong-Gallegos et al. [28] used the survey conducted by Gini et al. [29] that includes questions on general and neurodevelopmental neuromyths (i.e., pseudoscientific beliefs). It should be noted that, within the EUB research in Chile, there is ample literature on conspiracy beliefs compared to paranormal and pseudoscientific beliefs, the latter being the ones with the least scientific research to date.

In this scenario, where the relevance of studying the EUB is evident, the purpose of this work is to enrich the availability of tools for their study, inspired by the original proposal of Lobato et al. [2], because it is the most widely used in the literature, with the distinguishing of: 1) offer evidence of validity, according to the state of the art of psychometrics, for the interpretation of the scores in the local context; 2) have a wider context for use it, because the adapted content transcend local's idiosyncrasy; 3) provide a short scale for be included easily in all kind of studies; and 4) examine potential differences in EUB levels based on participants' gender, political orientation, and religious affiliation. Therefore, the present study aims to develop a brief scale, for its easy integration in batteries of studies, to evaluate the EUB in the Chilean population or equivalent.

## Materials and methods

### Participants

A total of 634 adults participated in the present study, of these, 30.1% (n = 191) resided in Arica, 27.9% (n = 177) in Santiago, 14.8% (n = 94) in Antofagasta, 14.6% (n = 93) in Talca and 12.6% (n = 80) in Punta Arenas. The mean age was 21.9 years (SD = 4.46). In terms of gender, 57.2% (n = 363) identified as female, 39.4% (n = 250) as male and 3.4% (n = 22) as non-binary or other. The majority of participants 93.9% (n = 596) were university students. Regarding religious affiliation 46. 8% (n = 289) identified as Catholic, Christian or affiliated with another religion, while 53.2% (n = 329) reported being agnostic, atheist or having no religion. Politically, 49.6% (n = 289) identified with the left, 8.2% (n = 48) with the center, 15.6% (n = 91) with the right and 26.6% (n = 155) reported being apolitical.

### Procedure

The original 37 items proposed by the Lobato et al. [2] were first translated and back-translated. Based on this version, 3 expert judges from the Universidad de Magallanes conducted a content analysis and defined 5 exclusion criteria: (1) items referencing culturally specific events (e.g., President John F. Kennedy was assassinated by Lee Harvey Oswald, who acted alone); (2) items potentially limited by age relevance (e.g., The musician Elvis Presley is dead); (3) items involving religious beliefs (e.g., Human beings have souls that continue to exist after the body dies); (4) items expressing hate

ideology (e.g., A person chooses to be homosexual, bisexual, or heterosexual), and (5) reverse-coded items in the pseudoscience dimension, as scientific beliefs are not necessarily the inverse of pseudoscientific ones (e.g., The beginning of the universe is best explained by the Big Bang Theory).

Some items were reworded for clarity to the target population (e.g., "Members of the US government were involved in the planning and execution of the events that happened on 11 September 2001" was revised to "There are agents of the US government who participated in the attack on the Twin Towers"). To address underrepresented in the conspiracy and pseudoscience dimensions, 2 new items were created for the conspiracy dimension (i.e., "There are secret groups of powerful people who make the most important decisions about how the world is managed at the international level" and "There are treatments that are effective for treating cancer but are hidden for economic purposes"), based on the definition of conspiratorial beliefs [22], including one health-related item. One additional item was created for the pseudoscience dimension (i.e., "It is necessary for an ordinary person to perform detoxification diets on a daily basis"), as such beliefs are commonly cited [30].

The resulting 22-item pilot scale was administered. Item analyses identified low-variability or low-homogeneity items, leading to the exclusion of 9 items from the paranormal dimension (e.g., Creatures popularly known as Bigfoot, the Loch Ness Monster, and/or the Chupacabra exist) and 1 item from the pseudoscience dimension (i.e., Childhood vaccines are one causal factor in the development of autism). For details on the item reduction process, see S1 Appendix.

The final version, used in this study, comprised 12 items evenly distributed across the 3 dimensions proposed by Lobato et al [2].

Data collection took place between August and December 2023. Five trained surveyors (one per city) supported the data collection process. Recruitment was conducted via posters. Interested participants signed a written informed consent form that outlined the study's purpose and guaranteed confidentiality and anonymity. The survey took approximately 25 minutes, and participants received a monetary. This study followed an instrumental approach, applying a battery of measures within a cross-sectional design. The Scientific Ethics Committee of the Universidad de Tarapacá granted ethical approval to this research, framed in the regular FONDECYT project n°1220664.

## Instruments

The Epistemically unwarranted beliefs scale (EUBS) was developed to assess individuals' adherence to such beliefs. The final scale includes three dimensions: pseudoscientific beliefs (e.g., Most humans only use about 10% of their brain), conspiracy beliefs (e.g., There are secret groups of powerful people who make the most important decisions about how the world is run on an international level) and paranormal beliefs (i.e., Astrology provides valuable information about how people are), with 4 items per dimension (12 items in total). Items were rated on a five-point Likert format (1 = strongly disagree; 5 = strongly agree), where higher scores indicate higher adherence.

The Cognitive reflection test (CRT), was based on the original version by Frederick [31] and CRT-2 by Thomson and Oppenheimer [32]. This test evaluates cognitive reflection, i.e., the ability of individuals to suppress automatic responses that seem obvious to find the correct answer. It consists of 7 items (e.g., A sheep farmer had 15 sheep. Except for 8, all of them died. How many live sheep did he have left?), were adapted to the local context, including references to local currency, culturally familiar scenarios and more accessible language.

## Analysis

Descriptive statistics, normality tests, corrected item-total correlations and reliability indices were first calculated. To evaluate internal structure validity, an Exploratory Structural Equation Modeling (ESEM) was conducted with TARGET rotation [33] using Weighted Least Squares Mean and Variance adjusted (WLSMV) estimation, which is robust for non-normal categorical data [34,35]. Reliability was assessed using Cronbach's alpha and McDonald's omega coefficients [36].

Next, Confirmatory Factor Analysis (CFA) was performed using WLSMV estimation for categorical variables [33] comparing a unidimensional and a bifactor model. To test measurement invariance across gender, multi-group CFA was conducted for the two largest gender groups (i.e., male and female), testing for metric and scalar invariance. A decrease of less than 0.010 in the Comparative Fit Index (CFI) was considered evidence of invariance [37].

Subsequently, validity based on relations with other variables was assessed via a structural equation model using polychoric correlations, with TARGET rotation and the WLSMV estimation, treating CRT as the independent variable, and a bivariate analysis using a Mixed ANOVA with Tukey HSD post-hoc test to examine differences in scores across gender, political and religious orientation. Gender was recoded into two categories (male and female), political orientation into four (left, center, right, apolitical), and religious affiliation into three (religious, non-religious, and atheist/agnostic). A significance threshold of $p < .001$ (0.1%) was applied.

Model fit was evaluate using standard indices: Tucker-Lewis Index (TLI), CFI, Root Mean Square Error of Approximation (RMSEA) with it is 90% Confidence Interval (90% CI). CFI and TLI values ≥0.95 and RMSEA ≤ .070 were considered indicators of adequate fit [38].

Descriptive statistics, reliability coefficients, item homogeneity indices, and mixed ANOVA were computed using Jamovi v2.0.0 [39], while ESEM and CFA were conducted using Mplus v8.2 [40].

## Results

### Evidence of validity based on internal structure

The 12-item ESEM model of the EUBS (Model 1; M1) demonstrated acceptable fit indicators according to recommendations in the literature [38] (see Table 1), however, several issues were identified: cross-loadings and low factor loadings for items theoretically assigned to a specific factor (i.e., PS4 and CO4), and one item that disproportionately loaded onto a single factor (i.e., PA4) (see Table 2). Based on these observations, a revised 9-items version of the EUBS was tested (Model 2; M2). The ESEM analysis of the second model showed improved factor clarity and reduced cross-loadings compared to M1. In addition, along with satisfactory fit indicators [38] (see Table 1).

Subsequently, two alternative models were tested using the 9-item version, a unidimensional model (Model 3; M3), which assumes a single latent factor underlying all items, and a bifactor model (Model 4; M4), which posits a general factor representing epistemically unwarranted beliefs, along with 3 specific factors – pseudoscientific, conspiracy and paranormal beliefs. Following the criteria by Hair et al.. [38], M4 showed a better fit than M3. Fig 1 presents the factor loadings of M4.

### Factorial invariance by gender

The differences in CFI between the metric and scalar models compared with the configural model, did not exceed the threshold of 0.10 (see Table 3), indicating invariance across gender [37]. This suggests that factor loadings and intercepts are equivalent for individuals identifying as female and male, meaning the 9 EUBS items are interpreted consistently across these groups.

**Table 1. ESEM and CFA fit indices for EUBS.**

|  | χ2 | df | χ²/ df | RMSEA | 90% CI | CFI | TLI | SRMR |
|---|---|---|---|---|---|---|---|---|
| M1 | 74.746* | 33 | 2.265 | .045 | [.031,.058] | .992 | .984 | .019 |
| M2 | 25.801* | 12 | 2.150 | .043 | [.019,.065] | .996 | .988 | .014 |
| M3 | 188.973* | 27 | 6.999 | .097 | [.084,.111] | .954 | .938 | .042 |
| M4 | 96.694* | 18 | 5.372 | .083 | [.067,.100] | .978 | .955 | .029 |

*$p < .001$; M1 = ESEM 12 items; M2 = ESEM 9 items; M3 = Unifactorial model; M4 = Bifactor model; RMSEA = Root Mean Square Error of Approximation; 90% CI = Confidence Interval; CFI = Comparative Fit Index; TLI = Tucker-Lewis Index; SRMR = Standardized Root Mean Square Residual.

**Table 2. Descriptive statistics and item-level factor loadings of M1 and M2.**

| | Descriptive statistics | | | | Factor loadings | | | | | |
| | M (SD) | S | K | SW | M1 | | | M2 | | |
| | | | | | PS | CO | PA | PS | CO | PA |
|---|---|---|---|---|---|---|---|---|---|---|
| PS1 | 2.53 (0.96) | .024 | −0.18 | .868 | **.549** | .116 | .079 | **.769** | .158 | −.047 |
| PS2 | 2.59 (1.48) | .300 | −1.40 | .840 | **.481** | .127 | .104 | **.396** | .065 | .221 |
| PS3 | 2.17 (1.03) | .125 | −0.61 | .858 | **.685** | .000 | .118 | **.550** | −.025 | .237 |
| PS4 | 2.64 (1.19) | .206 | −0.92 | .905 | **.301** | .283 | −.056 | | | |
| CO1 | 3.43 (1.11) | −.526 | −0.36 | .892 | .049 | **.737** | −.000 | .126 | **.813** | −.092 |
| CO2 | 2.97 (1.30) | −.056 | −1.08 | .905 | .204 | **.615** | −.058 | .160 | **.507** | .191 |
| CO3 | 3.31 (0.99) | −.241 | −0.03 | .893 | −.090 | **.452** | .237 | −.077 | **.427** | .252 |
| CO4 | 2.47 (1.20) | .357 | −0.76 | .886 | .304 | **.290** | .174 | | | |
| PA1 | 2.24 (1.19) | .510 | −0.84 | .854 | .352 | −.179 | **.617** | .312 | −.148 | **.600** |
| PA2 | 2.60 (1.32) | .225 | −1.17 | .882 | .102 | .221 | **.535** | .007 | .177 | **.660** |
| PA3 | 3.31 (1.22) | −.328 | −0.74 | .904 | −.060 | .350 | **.419** | −.059 | .317 | **.423** |
| PA4 | 2.76 (1.34) | .070 | −1.22 | .889 | −.200 | −.079 | **.999** | | | |
| CO | | | | | .513 | – | – | .473 | – | – |
| PA | | | | | .623 | .626 | – | .706 | .629 | – |

*p < .001; M1 = ESEM 12 items; M2 = ESEM 9 items; M = media; SD = Standard deviation; S = Asymmetry; K = Kurtosis; SW = Shapiro-Wilk test; PS = Pseudoscience; CO = Conspiracy; PA = Paranormal.

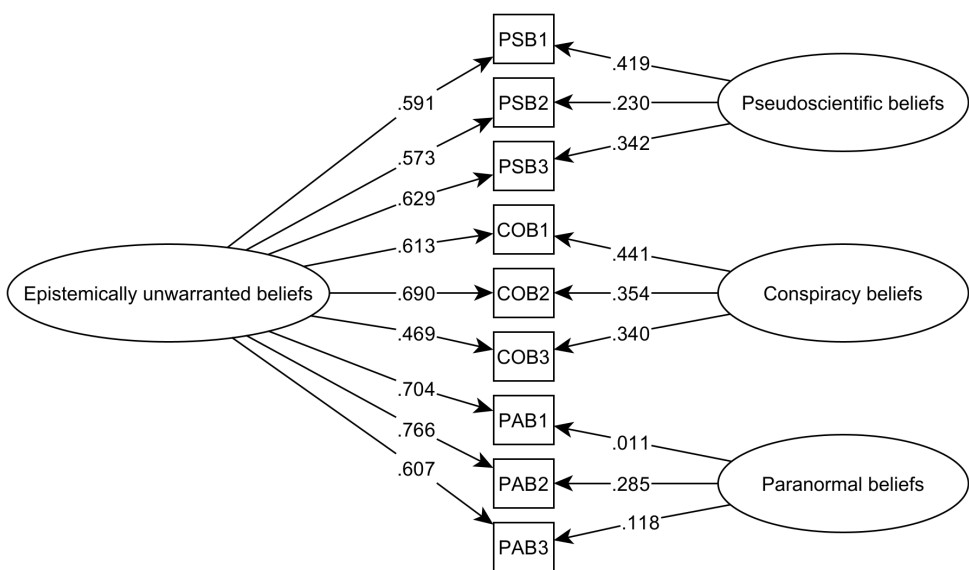

**Fig 1. Bifactor model, graphical representation.** In addition, the reduced 9-item epistemically unwarranted beliefs scale, demonstrated adequate internal consistency for the overall scale ($\omega = 0.833$; $\alpha = 0.837$), and for each of its dimensions: pseudoscientific beliefs ($\omega = 0.696$; $\alpha = 0.704$), conspiracy beliefs ($\omega = 0.696$; $\alpha = 0.704$) and paranormal beliefs ($\omega = 0.680$; $\alpha = 0.691$) [41].

**Table 3. Fit indexes for multi-group CFA of the EUBS by gender.**

|  | χ2 | df | χ2/ df | RMSEA | 90% CI | CFI | TLI | SRMR | CMs | ΔCFI |
|---|---|---|---|---|---|---|---|---|---|---|
| M5 | 149.507* | 48 | 3.114 | .083 | [.068,.098] | .857 | .786 | .055 |  |  |
| M6 | 154.760* | 54 | 2.866 | .078 | [.064,.093] | .858 | .811 | .058 | M6-M5 | .001 |
| M7 | 164.431* | 60 | 2.741 | .075 | [.062,.089] | .853 | .824 | .059 | M7-M6 | −.005 |

*p<.001; M5 = Configural invariance; M6 = Metric invariance; M6 = Scalar invariance; RMSEA = Root Mean Square Error of Approximation; 90% CI = Confidence Interval; CFI = Comparative Fit Index; TLI = Tucker-Lewis Index; SRMR = Standardized Root Mean Square Residual; CMs = Comparisons between models; ΔCFI = CFI differential.

## Evidence of validity based on the relationship with other variables

The impact of cognitive reflection test on the 9-item version of the EUBS was assessed using a Set-ESEM model (see Fig 2). This model demonstrated satisfactory fit indicators [39]: χ2(86) = 105.646; CFI = .995; TLI = .993; and RMSEA = .019 (confidence interval:.000 −.030), showing that it is an adequate representation of the observed relationships. According to the standardized effects, the 3 direct effects were statistically significant, showing a strong inverse effect ($\gamma$ = −.303, $p$ < .001 for pseudoscientific beliefs; $\gamma$ = −.177, $p$ < .001 for conspiracy beliefs; and $\gamma$ = −.545, $p$ < .001 for paranormal beliefs).

With regard to sociodemographic variables, the mixed ANOVA revealed significant main effects on all three belief domains: pseudoscientific beliefs, $F_{(17)}$ = 5.772, $p$ < .001; conspiracist beliefs, $F_{(17)}$ = 2.648, $p$ < .001; and paranormal beliefs, $F_{(17)}$ = 6.170, $p$ < .001. Tukey post-hoc comparisons showed significant difference in pseudoscientific beliefs based on religious orientation, with higher scores among individuals with religious affiliation compared to those without

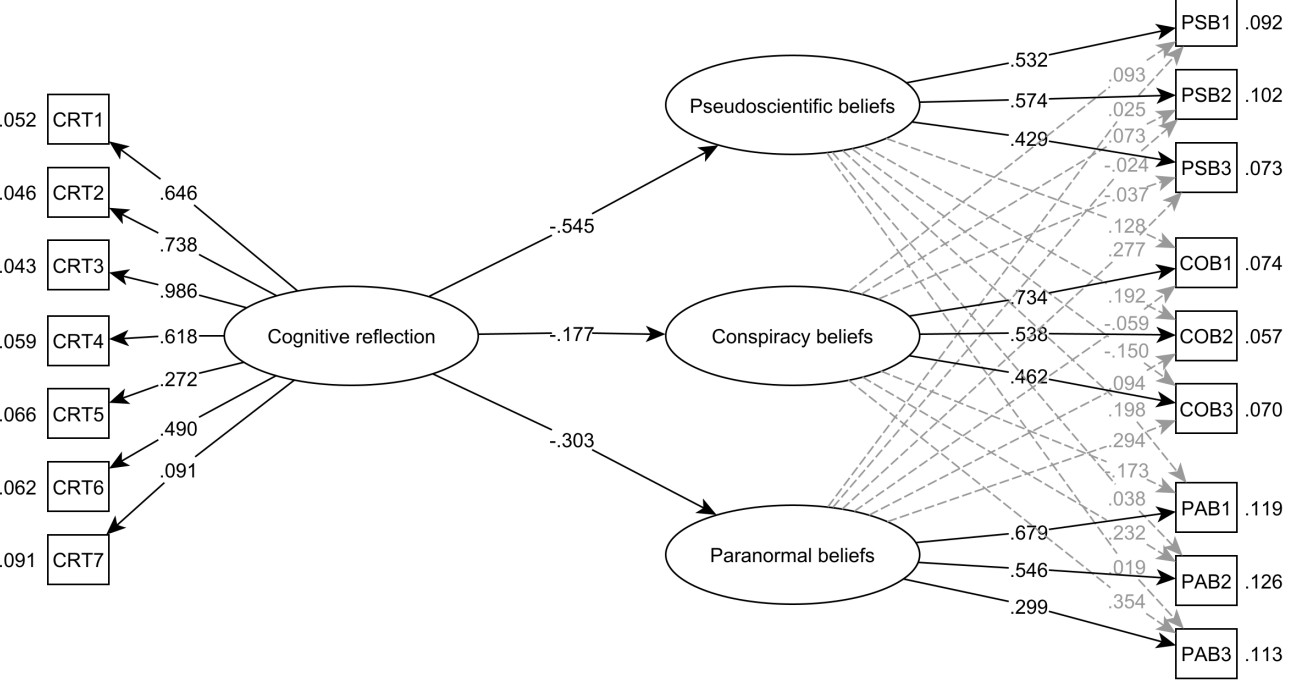

**Fig 2. Set-ESEM model, graphical representation of the relationships between Cognitive Reflection Test and EUBS.**

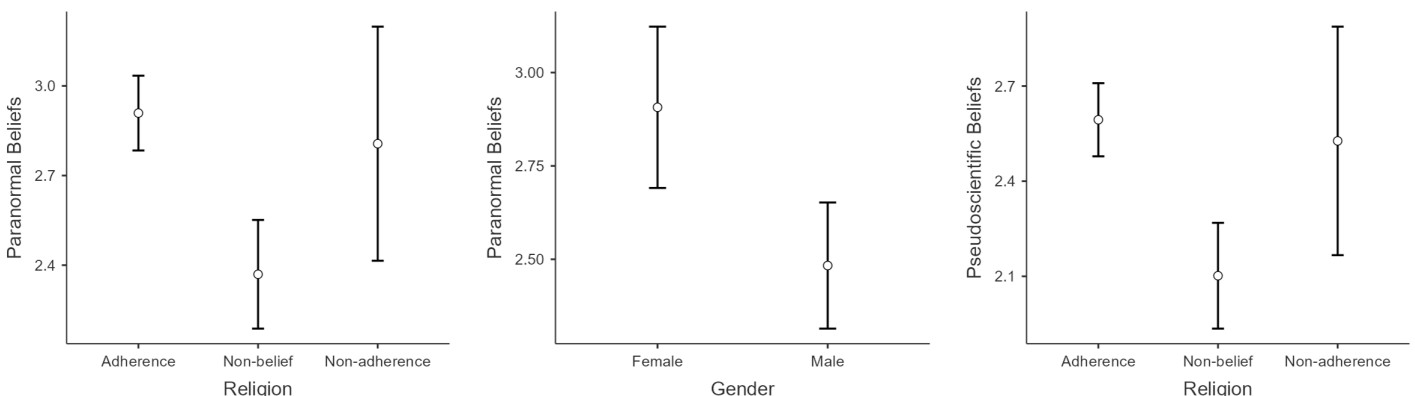

**Fig 3. Mean scores of pseudoscientific and paranormal beliefs by religious orientation and gender.**

belief in the divine (mean difference = 0.49, SE = 0.10; t(533) = 4.92, p < .001; d = 0.59) (see Fig 3). Regarding paranormal beliefs, significant differences were found by gender (female vs. male), with a mean difference of 0.42 (SE = 0.12; t(535) = 3.55, p < .001; d = 0.47), and by religious orientation (adherence to a religious belief vs. non-belief in the divine), with a mean difference of 0.54 (SE = 0.11; t(535) = 4.96, p < .001; d = 0.60) (see Fig 3).

No significant differences were observed in conspiracy beliefs between groups (for additional results, see S2 Appendix). Moreover, no significant interaction effects were found among the variables of gender, political and religion orientation.

## Discussion

The main aim of this study was to develop a brief scale that measures epistemologically unwarranted beliefs in an adult population. The Epistemically Unwarranted Beliefs Scale (EUBS) (see S3 Appendix) presented a stable structure composed of three factors and nine items with acceptable levels of internal consistency. In this line, the fit of the final model, the magnitude of the factor loadings and the cross-loadings corroborate the three-dimensional structure of the model. In addition, the estimates of the reliability coefficients, for each dimension and for the overall scale, show a satisfactory level of consistency.

The nine-item model also shows compatibility with a bifactor structure, i.e., it has a global factor, in this case, epistemologically unwarranted beliefs, and three specific factors that has some independence with the general factor, in this case the three beliefs studied (i.e., pseudoscientific, paranormal and conspiracy beliefs). Therefore, it is plausible to use the scale using its three specific and one general dimensions. These findings provide evidence for the structure of three related factors, i.e., pseudoscientific beliefs, paranormal beliefs, and conspiracy beliefs, in line with previous findings [1,2,22]. Furthermore, the nine-item model shown metric and scalar invariance of measurement between genders (i.e., women and men). This means that factor loadings and variability of dimensions were equivalent between women and men. This provides the opportunity to investigate adherence to epistemically unwarranted beliefs without distinction in the score interpretation between men and women.

Regarding validity evidence based on the relationship with other variables, CRT was found to correlate negatively with the EUBS dimensions, aligning with the theoretically expected direction and corroborating previous studies [22,42,43]. In other words, people with epistemically unwarranted beliefs present lower capacity for cognitive reflection, understood as the ability to inhibit intuitive responses and reach the correct solution. However, these results should be interpreted

with caution, as recent evidence suggests that CRT performance may also reflect the ability to provide correct intuitive responses, particularly when individuals possess well-internalized logical knowledge [44,45].

Also, the present findings are consistent with previous research showing that epistemically unwarranted beliefs (EUB) vary according to sociodemographic and ideological factors. Significant differences were found in pseudoscientific and paranormal beliefs related to religious orientation and gender, supporting earlier evidence of higher EUB endorsement among religious individuals and females [1,16]. However, no significant differences emerged for conspiracist beliefs, suggesting that other variables might better explain this domain. Future research should explore additional psychological and social factors to better understand the complexity behind the endorsement of unfounded beliefs.

Considering the significant impact of the EUB on people's health behaviors [5,6], the incorporation of this scale in health or education settings could be beneficial. The information provided by this scale could help to identify persons more likely to adhere to these beliefs, and to implement interventions to help mitigate the adverse effects of believing in paranormal, conspiracy, and pseudoscientific theories. For example, Dyer and Hall [13], in a university population, conducted a critical thinking intervention that significantly reduced pseudoscientific beliefs. In addition, its application in social psychology, public safety and media would allow understanding how EUBs influence institutional distrust, risk perception and the dissemination of misinformation. This would facilitate the design of strategies to promote critical thinking and strengthen citizen trust.

The main limitations of this study refer to the representativeness of the sample. On the one hand, the participants in our study were recruited through a non-probabilistic sample, which could affect the accuracy of representativeness, however, it should be considered that the participants belong to five different regions of Chile, covering different geographical areas (i.e., north, center and south). On the other hand, most of the participants were university students, which could bias our results to a population with specific characteristics. Consequently, it is recommended that future psychometric studies using this instrument be extended to different age groups (e.g., adolescents, older adults) with different sociodemographic variables (e.g., educational and socioeconomic levels).

Finally, future research should attempt to validate this scale in other Spanish-speaking countries and expand its use in different cultures and contexts (e.g., health, education, etc.). Furthermore, given that this instrument is a brief scale, it could be useful to incorporate it into larger psychometric studies and compare its results with those of other variables that correlate with epistemologically suspect beliefs (e.g., political leanings, adherence to misinformation) [23,25,46].

## Supporting information

**S1 Appendix. Item reduction process.**
(DOCX)

**S2 Appendix. Multiple post hoc comparisons.**
(DOCX)

**S3 Appendix. Epistemically unwarranted beliefs scale.**
(DOCX)

## Author contributions

**Conceptualization:** Rodrigo Ferrer-Urbina, Herman Elgueta, Marcos Carmona-Halty, Geraldy Sepúlveda-Páez.

**Data curation:** Rodrigo Ferrer-Urbina, Geraldy Sepúlveda-Páez, Karina Alarcón-Castillo.

**Formal analysis:** Rodrigo Ferrer-Urbina.

**Funding acquisition:** Rodrigo Ferrer-Urbina.

**Investigation:** Rodrigo Ferrer-Urbina, Herman Elgueta, Marcos Carmona-Halty, Geraldy Sepúlveda-Páez, Karina Alarcón-Castillo.

**Methodology:** Rodrigo Ferrer-Urbina, Herman Elgueta, Geraldy Sepúlveda-Páez.

**Project administration:** Rodrigo Ferrer-Urbina.

**Resources:** Rodrigo Ferrer-Urbina.

**Software:** Rodrigo Ferrer-Urbina.

**Supervision:** Rodrigo Ferrer-Urbina, Karina Alarcón-Castillo.

**Validation:** Rodrigo Ferrer-Urbina, Herman Elgueta, Geraldy Sepúlveda-Páez, Karina Alarcón-Castillo.

**Visualization:** Rodrigo Ferrer-Urbina, Karina Alarcón-Castillo.

**Writing – original draft:** Rodrigo Ferrer-Urbina, Karina Alarcón-Castillo.

**Writing – review & editing:** Rodrigo Ferrer-Urbina, Herman Elgueta, Marcos Carmona-Halty, Geraldy Sepúlveda-Páez, Karina Alarcón-Castillo.

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
