## [Decision Letter · Decision Letter 0]

19 Jun 2025

PONE-D-25-14825Epistemically Unwarranted Beliefs Scale, development and evidence of validity in the Chilean population.

PLOS ONE

Dear Dr. Alarcón,

Thank you for submitting your manuscript to PLOS ONE. After careful consideration, we feel that it has merit but does not fully meet PLOS ONE’s publication criteria as it currently stands. Therefore, we invite you to submit a revised version of the manuscript that addresses the points raised during the review process.

A rebuttal letter that responds to each point raised by the academic editor and reviewer(s). You should upload this letter as a separate file labeled 'Response to Reviewers'.A marked-up copy of your manuscript that highlights changes made to the original version. You should upload this as a separate file labeled 'Revised Manuscript with Track Changes'.An unmarked version of your revised paper without tracked changes. You should upload this as a separate file labeled 'Manuscript'.Please ensure that the revised submission includes a copy of the full instrument and instructions as a Supporting Information file or provide a link if it is available through an online repository, in alignment with PLOS ONE requirements stating that the questionnaire or scale must be openly available under a license no more restrictive than CC BY. By uploading a copy of the questionnaire you have used for this study, you are confirming that you have obtained all the necessary permissions to publish the questionnaire CC BY and have included all proper attributions in your manuscript. For more detail, please see here: https://journals.plos.org/plosone/s/licenses-and-copyright . If you are unable to obtain the necessary permissions, please indicate this.

We look forward to receiving your revised manuscript.

Kind regards,

Fernando Blanco

Academic Editor

PLOS ONE

Journal Requirements:

Additional Editor Comments (if provided):

Dear author

First of all, I would like to apologise for the long time this review process has taken. It was particularly challenging to find suitable reviewers, but I have finally obtained two expert reports that I am quite satisfied with. The reviewers come from slightly different fields, which allows their feedback to complement each other well.

Both reviewers are generally positive about the manuscript, but they offer different perspectives and suggestions for improvement, which I believe will be valuable for a major revision. I summarise the main comments below:

The items removed from the full instrument should be included in the manuscript as the reviewer indicates.In the Methods section, please specify the type of study or methodology used (e.g., cross-sectional, descriptive, etc.) as well as the sampling method.One reviewer suggests using a scree plot to convey the results of the exploratory factor analysis.In the Discussion section, a more detailed explanation about the sociodemographic variables would be a good addition (see the reviewer’s comment).I also encourage to follow the reviewer’s advice to discuss alternative interpretations of the CRT, which I find particularly insightful in the context of this study.Although the manuscript reads quite smoothly overall, a careful review of the writing is advisable to ensure clarity and ease of understanding. I have noticed that the methodological and results sections are a bit harder to follow.

Reviewers' comments:

Reviewer's Responses to Questions

**Comments to the Author**

1. Is the manuscript technically sound, and do the data support the conclusions?

Reviewer #1: Yes

Reviewer #2: Yes

2. Has the statistical analysis been performed appropriately and rigorously? 

Reviewer #1: Yes

Reviewer #2: Yes

3. Have the authors made all data underlying the findings in their manuscript fully available?

Reviewer #1: Yes

Reviewer #2: Yes

4. Is the manuscript presented in an intelligible fashion and written in standard English?

Reviewer #1: Yes

Reviewer #2: Yes

5. Review Comments to the Author

Reviewer #1: Thank you for submitting your manuscript entitled "Epistemically Unwarranted Beliefs Scale, development and evidence of validity in the Chilean population." The main claim of the paper is the development and validation of the Epistemically Unwarranted Beliefs Scale (EUBS), a brief, psychometrically sound instrument designed to measure pseudoscientific, paranormal, and conspiracy beliefs in the Chilean population. The study demonstrates that the scale has a clear three-factor structure, good internal consistency, gender invariance, and expected negative correlations with cognitive reflection. This contribution is significant for the discipline as it fills a critical gap by providing a culturally adapted tool for assessing epistemically unwarranted beliefs in Latin America, enabling more accurate research on the cognitive and social correlates of such beliefs in diverse populations.

This is a timely and relevant contribution to the field of psychological assessment, especially considering the impact of unwarranted beliefs on public health and misinformation. The development of a culturally adapted, brief, and psychometrically sound instrument such as the Epistemically Unwarranted Beliefs Scale (EUBS) is a meaningful advancement, particularly in Latin American contexts where validated tools are scarce.

The manuscript is well-organized and generally clear, and the methodological rigor is commendable. I appreciate the use of Exploratory Structural Equation Modeling (ESEM), confirmatory factor analysis, and measurement invariance testing. The inclusion of cognitive reflection as a validity criterion is appropriate and theoretically grounded.

That said, I would like to offer the following suggestions for improvement:

While the justification for developing a culturally adapted scale is strong, the manuscript would benefit from deeper engagement with cross-cultural measurement theory. Specifically, consider expanding the discussion of how sociodemographic variables might affect EUB manifestations and their measurement. I suggest to consult previous research, such as the study by Torres et al. (2023; https://www.nature.com/articles/s41599-023-01681-3), where a series of cognitive and sociodemographic correlates of the different unwarranted beliefs (mainly pseudoscientific ones) are addressed. So an appropriate hypothesis regarding sociodemographic variables could be formulated and your results could be compared with previous research conducted with other population.

The sample is heavily composed of university students (93.9%), which may limit generalizability. This limitation is acknowledged in the discussion, but I encourage the authors to explicitly mention this in the abstract as well. Additionally, considering the rich sociodemographic data collected and linking to the previous consideration, a more detailed analysis of associations between EUB and variables such as political orientation or religious belief could have enriched the findings.

Line 87: I recommend to reword this statement since it is not fully clear the direction of the correlation between variables. Is it positive (higher conspiracy beliefs, higher effectiveness of COVID-19 vaccines) or negative (higher conspiracy beliefs, lower effectiveness of the COVID-19 vaccines)?

The rationale behind the inclusion or exclusion of specific items could be improved. A brief justification for the new items created by the authors would strengthen the transparency of the development process:

• Line 125: I recommend to include an example of the mentioned rewording.

• Lines 127-132: a brief explanation of on what basis were these items created is desirable.

Interpretation of the Cognitive Reflection Test (CRT): While CRT is an appropriate external correlate, their interpretation should be revised. Recent work (e.g., Bago & De Neys, 2019; Šrol & De Neys, 2021) suggests that CRT performance may arise through intuitive processes rather than through deliberate analytical reasoning. This challenges the traditional interpretation of CRT performance as primarily reflecting the ability to override intuitive but incorrect responses via deliberation. Instead, these studies point to an alternative account: CRT scores may reflect an individual’s sensitivity to conflict between heuristic intuitions and logical intuitions.

Also, check the statement on lines 264 and 265 taking into account Bago and De Neys (2019) and Šrol and De Neys (2021) studies.

https://www.tandfonline.com/doi/full/10.1080/13546783.2018.1507949

https://www.tandfonline.com/doi/full/10.1080/13546783.2019.1708793

Minor Issues:

• While the manuscript is readable, there are some typo errors that must be addressed:

- Line 35: “communication medias” should be “communication media”.

- Line 179: “model restrictyions” should be “model restrictions”.

- Line 2019: “pseudoscience belief” shoud be “pseudoscientific beliefs” or “beliefs in pseudoscience”.

A thorough language review would improve clarity and professionalism.

• Check the reference in line 81, as it should be 38 instead of 20. Likewise, check that the reference of the line 82 is correct.

• Please ensure consistent and correct use of decimal points: e.g., use “0.833” rather than “0,833” (lines 218-220).

In summary, this study addresses a critical gap in the literature by offering a brief, reliable, and valid measure of epistemically unwarranted beliefs in the Chilean context. With revisions aimed at strengthening the theoretical rationale and methodological transparency, this manuscript has the potential to make a valuable contribution to psychological assessment and cross-cultural research on belief systems.

Reviewer #2: It would also be useful to include the final version of the instrument with instructions for use. Regarding the item reduction process, greater transparency is suggested, including an appendix documenting the items that were eliminated.

6. PLOS authors have the option to publish the peer review history of their article (what does this mean? ). If published, this will include your full peer review and any attached files.

**Do you want your identity to be public for this peer review?** For information about this choice, including consent withdrawal, please see our Privacy Policy .

Reviewer #1: No

Reviewer #2: No

---

## [Author Response · Author response to Decision Letter 1]

30 Aug 2025

Editor:

Comment 1:

The items removed from the full instrument should be included in the manuscript as the reviewer indicates.

Response:

We added the removed items according to both reviewers' suggestions. For Reviewer 1, they were included in lines 146–149, 139–142, and 461–473; for Reviewer 2, in lines 461–473.

Comment 2:

In the Methods section, please specify the type of study or methodology used (e.g., cross-sectional, descriptive, etc.) as well as the sampling method.

Response:

This information was added (lines 162–163).

Comment 3:

One reviewer suggests using a scree plot to convey the results of the exploratory factor analysis.

Response:

We did not include a scree plot because we conducted an Exploratory Structural Equation Modeling (ESEM) instead of a traditional exploratory factor analysis. However, we are open to including it if deemed necessary.

Comment 4:

In the Discussion section, a more detailed explanation about the sociodemographic variables would be a good addition (see the reviewer’s comment).

Response:

We incorporated a comparative analysis with prior studies in the Discussion section (lines 305–312).

Comment 5:

I also encourage to follow the reviewer’s advice to discuss alternative interpretations of the CRT, which I find particularly insightful in the context of this study.

Response:

New information based on the studies suggested by Reviewer 1 was added (lines 301–304).

Comment 6:

Although the manuscript reads quite smoothly overall, a careful review of the writing is advisable to ensure clarity and ease of understanding. I have noticed that the methodological and results sections are a bit harder to follow.

Response:

The manuscript was revised for improved clarity and readability.

Reviewer #1:

Comment 1:

While the justification for developing a culturally adapted scale is strong, the manuscript would benefit from deeper engagement with cross-cultural measurement theory. Specifically, consider expanding the discussion of how sociodemographic variables might affect EUB manifestations and their measurement. I suggest to consult previous research, such as the study by Torres et al. (2023; https://www.nature.com/articles/s41599-023-01681-3), where a series of cognitive and sociodemographic correlates of the different unwarranted beliefs (mainly pseudoscientific ones) are addressed. So an appropriate hypothesis regarding sociodemographic variables could be formulated and your results could be compared with previous research conducted with other population.

Response:

We followed your suggestion by conducting mixed ANOVA analyses. A new theoretical section was added on sociodemographic variables related to EUB (gender, political and religious orientation) (lines 53–61 and 112–113). We also detailed the analyses in the Methods (195–200), presented results (261–274), added a figure (Figure 3), and included a supplementary table (Appendix 2). Comparisons with previous findings were discussed (lines 305–312).

Comment 2:

The sample is heavily composed of university students (93.9%), which may limit generalizability. This limitation is acknowledged in the discussion, but I encourage the authors to explicitly mention this in the abstract as well. Additionally, considering the rich sociodemographic data collected and linking to the previous consideration, a more detailed analysis of associations between EUB and variables such as political orientation or religious belief could have enriched the findings.

Response:

The abstract now reflects the student sample (lines 21–22) and the sociodemographic analyses (lines 26–27). The detailed analysis is described in the response to Comment 1.

Comment 3:

Line 87: I recommend to reword this statement since it is not fully clear the direction of the correlation between variables. Is it positive (higher conspiracy beliefs, higher effectiveness of COVID-19 vaccines) or negative (higher conspiracy beliefs, lower effectiveness of the COVID-19 vaccines)?

Response:

The statement was revised for clarity (lines 100–101).

Comment 4:

The rationale behind the inclusion or exclusion of specific items could be improved. A brief justification for the new items created by the authors would strengthen the transparency of the development process:

• Line 125: I recommend to include an example of the mentioned rewording.

• Lines 127-132: a brief explanation of on what basis were these items created is desirable.

Response:

Justifications were included (lines 146–149), a supplementary table for exclusions was added (lines 461–473), and a rewording example was included (lines 139–142).

Comment 5:

Interpretation of the Cognitive Reflection Test (CRT): While CRT is an appropriate external correlate, their interpretation should be revised. Recent work (e.g., Bago & De Neys, 2019; Šrol & De Neys, 2021) suggests that CRT performance may arise through intuitive processes rather than through deliberate analytical reasoning. This challenges the traditional interpretation of CRT performance as primarily reflecting the ability to override intuitive but incorrect responses via deliberation. Instead, these studies point to an alternative account: CRT scores may reflect an individual’s sensitivity to conflict between heuristic intuitions and logical intuitions. Also, check the statement on lines 264 and 265 taking into account Bago and De Neys (2019) and Šrol and De Neys (2021) studies.

https://www.tandfonline.com/doi/full/10.1080/13546783.2018.1507949

https://www.tandfonline.com/doi/full/10.1080/13546783.2019.1708793

Response:

We incorporated recent theoretical perspectives to encourage cautious interpretation (lines 301–304).

Comment 6:

While the manuscript is readable, there are some typo errors that must be addressed:

- Line 35: “communication medias” should be “communication media”.

- Line 179: “model restrictyions” should be “model restrictions”.

- Line 2019: “pseudoscience belief” shoud be “pseudoscientific beliefs” or “beliefs in pseudoscience”.

A thorough language review would improve clarity and professionalism.

Response:

All issues were addressed: line 35 (corrected in 39), line 179 (removed), line 201 (corrected in 236).

Comment 8:

Check the reference in line 81, as it should be 38 instead of 20. Likewise, check that the reference of the line 82 is correct.

Response:

References were corrected (lines 94–95).

Comment 9:

Please ensure consistent and correct use of decimal points: e.g., use “0.833” rather than “0,833” (lines 218-220).

Response:

This was corrected (lines 235–237).

Reviewer #2:

Comment 1:

It would also be useful to include the final version of the instrument with instructions for use. Regarding the item reduction process, greater transparency is suggested, including an appendix documenting the items that were eliminated.

Response:

The final instrument and instructions were added in S3 Appendix (lines 477–482), and item reduction was documented in S1 Appendix (lines 461–473).

---

## [Decision Letter · Decision Letter 1]

21 Sep 2025

Epistemically Unwarranted Beliefs Scale, development and evidence of validity in the Chilean population.

PONE-D-25-14825R1

Dear Dr. Alarcón,

We’re pleased to inform you that your manuscript has been judged scientifically suitable for publication and will be formally accepted for publication once it meets all outstanding technical requirements.

Kind regards,

Fernando Blanco

Academic Editor

PLOS ONE

Additional Editor Comments (optional):

Reviewer #1:

Reviewer #2:

Reviewers' comments: 

Reviewer's Responses to Questions

**Comments to the Author**

1. If the authors have adequately addressed your comments raised in a previous round of review and you feel that this manuscript is now acceptable for publication, you may indicate that here to bypass the “Comments to the Author” section, enter your conflict of interest statement in the “Confidential to Editor” section, and submit your "Accept" recommendation.

Reviewer #1: All comments have been addressed

Reviewer #2: All comments have been addressed

2. Is the manuscript technically sound, and do the data support the conclusions?

Reviewer #1: Yes

Reviewer #2: Yes

3. Has the statistical analysis been performed appropriately and rigorously? 

Reviewer #1: Yes

Reviewer #2: (No Response)

4. Have the authors made all data underlying the findings in their manuscript fully available?

Reviewer #1: Yes

Reviewer #2: Yes

5. Is the manuscript presented in an intelligible fashion and written in standard English?

Reviewer #1: Yes

Reviewer #2: Yes

6. Review Comments to the Author

Reviewer #1: Thank you for carefully addressing the comments from my previous peer review and submitting the revised manuscript. The revisions have notably strengthened the discussion of the findings, resulting in a more accurate and well-presented paper.

I therefore recommend the manuscript for acceptance and publication. I appreciate your thoughtful efforts in revising the work and responding to the feedback provided.

Reviewer #2: The authors have addressed the reviewers' comments, and I consider that the article is ready for publication.

7. PLOS authors have the option to publish the peer review history of their article (what does this mean? ). If published, this will include your full peer review and any attached files.

**Do you want your identity to be public for this peer review?** For information about this choice, including consent withdrawal, please see our Privacy Policy .

Reviewer #1: No

Reviewer #2: No

---

## [Editor Report · Acceptance letter]

PONE-D-25-14825R1

PLOS ONE

Dear Dr. Alarcón-Castillo,

I'm pleased to inform you that your manuscript has been deemed suitable for publication in PLOS ONE. Congratulations! Your manuscript is now being handed over to our production team.

Kind regards,

on behalf of

Dr. Fernando Blanco

Academic Editor

PLOS ONE